# VERCASM-CPS: Vulnerability Analysis and Cyber Risk Assessment for Cyber-Physical Systems

**Bradley Northern †, Trey Burks †, Marlana Hatcher, Michael Rogers and Denis Ulybyshev ***

Department of Computer Science, Tennessee Technological University, Cookeville, TN 38505, USA;
banorthern42@tntech.edu (B.N.); tmburks42@tntech.edu (T.B.); mhatcher42@tntech.edu (M.H.);
mrogers@tntech.edu (M.R.)
* Correspondence: dulybyshev@tntech.edu; Tel.: +1-931-372-6127
† These authors contributed equally to this work.

**Abstract:** Since Cyber-Physical Systems (CPS) are widely used in critical infrastructures, it is essential to protect their assets from cyber attacks to increase the level of security, safety and trustworthiness, prevent failure developments, and minimize losses. It is necessary to analyze the CPS configuration in an automatic mode to detect the most vulnerable CPS components and reconfigure or replace them promptly. In this paper, we present a methodology to determine the most secure CPS configuration by using a public database of cyber vulnerabilities to identify the most secure CPS components. We also integrate the CPS cyber risk analysis with a Controlled Moving Target Defense, which either replaces the vulnerable CPS components or re-configures the CPS to harden it, while the vulnerable components are being replaced. Our solution helps to design a more secure CPS by updating the configuration of existing CPS to make them more resilient against cyber attacks. In this paper, we will compare cyber risk scores for different CPS configurations and show that the Windows® 10 build 20H2 operating system is more secure than Linux Ubuntu® 20.04, while Red Hat® Enterprise® Linux is the most secure in some system configurations.

**Keywords:** cyber-physical systems; industrial control systems; data privacy; moving target defense; cyber-risk score

## 1. Introduction

Software and hardware used in CPS still have a substantial amount of vulnerabilities that can be potentially exploited by attackers. These vulnerabilities may occur because existing CPS software and hardware are not up-to-date or they are not properly configured. Furthermore, the old generation software, hardware and network protocols, such as MODBUS®, were designed without cybersecurity in mind [1]. A recent cyber attack on the water treatment facility in Florida changed "the level of sodium hydroxide, more commonly known as lye, in the water from 100 parts per million to 11,100 parts per million" [2]. This case shows that the consequences of cyber attacks on critical infrastructures might be very serious.

New vulnerabilities appear in the public database of Common Vulnerabilities and Exposures (CVE) almost every day. According to this CVE database hosted at cve.mitre.org, as of 4 August 2021 there are 1126 registered vulnerabilities in Microsoft® Windows® (this paper "VERCASM-CPS: Vulnerability Analysis and Cyber Risk Assessment for Cyber-Physical Systems" is an independent publication and is neither affiliated with, nor authorized, sponsored, or approved by, Microsoft Corporation®) 10 operating system, which is often used in Industrial Control Systems that are a subset of CPS. There are 21 vulnerabilities in the "Ignition"® Supervisory Control and Data Acquisition (SCADA) system. For example, the vulnerability CVE-2020-12004 allows an attacker to obtain sensitive information due to the lack of a proper authentication on the "Ignition"® 8 Gateway (versions prior to 8.0.10) and "Ignition"® 7 Gateway (versions prior to 7.9.14). CVE is a sponsored

development by the U.S. Department of Homeland Security (DHS), and The Cybersecurity and Infrastructure Security Agency (CISA). Since large-scale CPS may contain hundreds of software and hardware components, it is necessary to run the vulnerability analysis in an automatic mode and reconfigure or replace vulnerable CPS components promptly to make the CPS more resilient against cyber attacks. CPS reconfiguration is an important component of any security policy.

Work has been done to determine the CVE-based cyber risk associated with vulnerable software and hardware. However, most of this work has been applied to commodity systems, and none of it takes into account the need for continuous reconfiguration of CPS. In this paper, we propose a novel VulnERability and Cyber risk ASsessMent for Cyber-Physical Systems (VERCASM-CPS) methodology to determine and build the most secure configuration of CPS, relying on CVE-based cyber risk evaluation of its separate software and hardware components. Furthermore, we designed the mechanism to integrate the cyber risk evaluation engine with a Controlled Moving Target Defense (CMTD) in order to reconfigure or replace vulnerable CPS components in an automatic or manual modes. This allows to harden the CPS, reduce its cyber risks, and make the CPS more resilient against cyber attacks. The novelty of our solution includes the following:

1. CVE-based methodology to evaluate the Common Vulnerability Scoring System (CVSS) base score of each separate CPS component, including software and hardware components. This is used to compute the total CPS cyber risk score of an entire system;
2. Software framework for system administrators, CPS designers and operators to determine in automatic continuous mode the most secure CPS configuration for newly designed CPS and components to reconfigure or replace for existing CPS;
3. Sensitivity analysis ("What If" functionality) which allows to evaluate how replacing one or several CPS components would affect the total CPS cyber risk score;
4. Controlled Moving Target Defense to reconfigure or replace the most vulnerable CPS components in automatic or manual modes, based on the current state of the public CVE database.

The rest of the paper is organized as follows. In Section 2, we present the overview of the related work. In Section 3, we discuss the core design of our methodology. Section 4 contains evaluation and experimental results. Section 5 concludes the paper and Section 6 discusses the future plans to extend our work.

## 2. Related Work

Aksu et al. [3] presented a methodology that uses CVE scores to assess cyber risk in IT systems. Their approach incorporates many facets of the CVE scores given by NIST, and uses them to calculate a numerical product of probability of vulnerability exploitation and impact of the exploitation to determine the risk of attack based on the vulnerabilities. The approach takes into account threat sources and the chance of the attack occurring to estimate the risk. However, anticipating threat capability and motivation to come to a quantifiable measurement can lead to an inaccurate estimation unless the attacker is known to the victim in advance and/or prior extensive knowledge of the attacker is readily available. While CVE scores are meant to provide a measure of severity of the attack should it happen, it is unclear if using the metrics from the CVE score calculation would lead to a correct estimation of exploitation likelihood. They also dismiss certain vulnerabilities due to considering Intrusion Detection Systems (IDS) or Intrusion Prevention Systems (IPS) that may be in place, but do not account for the failure of these systems, time between intrusions and discovery for which the vulnerabilities could be exploited, or the ability of attackers bypassing these systems. Lastly, in contrast with our approach, there are no suggestions for mitigation strategies after the risk is calculated.

Spring et al. [4] from Carnegie Mellon University presented a paper explaining why the Common Vulnerability Scoring System (CVSS) [5] scores need to be more justified and transparent. They question some of the aspects of the CVSS calculation, such as the type of measurement and the translation of that measurement into numerical measure.

They wonder how the importance of each metric was selected and how they are weighted. Spring et al. remark that the CVSS scores indicate severity of a vulnerability, but the security community may need a different type of indication, namely risk imposed or needed response time, rather than solely the severity. They touch on the areas of dissatisfaction that have been discussed in the cybersecurity community for over 10 years, for instance, the operational scoring issues. Lastly in the paper, the authors suggest methods and practices to refine CVSS scores, such as user and scoring studies as well as a specific rubric to follow for scoring to occur. However, they offer no example formula to create or represent the scores, nor do they suggest a specific rubric to follow to score any software.

Chang et al. [6] investigated the overall trends of CVEs over the course of 3 years (2007–2010) in an effort to assist in the focusing of preventative and mitigation tactics. The researchers chose to examine the vulnerability types that encompassed the largest amount of CVEs, which include authentication problem, buffer overflow, cryptographic error, and Cross-Site Request Forgery. The data collection resulted in over 10,000 CVEs or nearly 50% of all reported during the time period. The authors found that the number of vulnerabilities had decreased, the severity of them had moved to lower levels, and the majority of them could be exploited through accessing the system's network without being checked for authentication at any point during the access. However, this work does not provide immediate tactics to mitigate vulnerabilities.

Tripathi and Singh [7] made an effort to improve response to vulnerabilities by prioritising them by category through developed security metrics. They retrieved information from the National Vulnerability Database (NVD) from April 2010 to May 2011 which categorizes the vulnerabilities into 23 classes. They chose this time frame to account for the largest amount of time it takes to employ fixes. The authors calculated a security score for each of the vulnerability classes using CVSS scores and two factors they deemed important to the impact on severity, namely the availability of the patches and how long ago the vulnerability was initially reported. These are taken into consideration because as time passes from the initial report, the likelihood of the existence and employment of fixes is likely to increase. The scores are then calculated for two divisions of the vulnerabilities, those with patches and those without, and combined into a weighted sum. Tripathi and Singh [7] found that the top three vulnerability categories are buffer errors, input validation, and resource management errors. Our methodology does not categorize vulnerabilities, but it continuously evaluates the cyber risk score for separate CPS components and the total CPS cyber risk score. Furthermore, our solution supports automatic replacement or reconfiguration of vulnerable CPS components.

Allodi and Massacci [8] created their own databases in an effort to bridge gaps in the NVD and EnterpriseDB (EDB) for cyber attacks in the wild. The resulting databases are much smaller than those of NVD and EDB, and indicate different results. One of the created databases contains vulnerabilities that are included in Exploit Kits that are sold on the black market trade. The second is built to collect and store information about the particular attacks the researchers have interest in. The database that includes Exploit Kits may not be representative of actual attacks happening through the kits. These kits might be untrusted as they are from malicious sources. The information collection database is derived from another database, Symantec, and therefore is entirely dependent on it to pinpoint threats. The authors also explored the sensitivity of CVSS scores and report that the scores are not a good indication of the actual exploitation of vulnerabilities reported. However, CVSS are supposed to be an indication of the severity of the attack should it be exploited, not if it will be exploited.

A company, Tenable® [9], has produced a proprietary software (Tenable® Lumin) that gathers necessary information from multiple sources to assess the vulnerability and cyber risk of software [10].

The Cyber Exposure Score (CES) is calculated using two individual scores dubbed Vulnerability Priority Rating (VPR), and Asset Criticality Rating (ACR). The VPR score is a machine-generated number obtained from Tenable®'s machine learning algorithm. It

is claimed that the VPR is calculated faster and in a way that prioritises the threat more than using CVSS alone. The ACR score is a user-influenced number that indicates the importance of an asset to the particular user. The Tenable® Lumin software has its own set of rules to assign the ACR score if the user does not look to employ that part of the software.

While the general mathematics and information are very straight forward, the machine learning algorithm, data sources, and anything there within are kept rightfully private by Tenable®. They provide users with a list of recommended actions to follow in order to reduce their CES, which can consist of updates and fixes that may or may not reduce the risk score depending on reports of the solution over time, rather than mitigation by switching to another software component that is considered less of a risk without fixes. The work by the Tenable® Lumin software relies on the geometric mean. They explain that this was chosen to ensure pieces of software with few vulnerabilities are scored closer to a zero value. This would seem to undervalue threats from components with few but very critical vulnerabilities. This software looks at the current setup of components and calculates the proper risk scores. The software would have to be reapplied after a component change. Our solution, in contrast, supports running a separate "What If" procedure to see the change in the total risk score for the system without actually changing the component. Furthermore, our solution supports automatic reconfiguration or replacement of vulnerable software components.

The software tool called "Vulners" [11] can perform the vulnerability audit for Linux systems and networks. In contrast with this tool, we compute the cyber risk score for each software component and the entire system. Furthermore, our solution can evaluate the cyber risk score for non-Linux software, as well as for the hardware. Finally, our solution is integrated with the CMTD, which replaces the vulnerable components and hardens the system while the reconfiguration is in process.

Schlette et al. [12] proposed security enumerations that can "cover various Cyber Threat Intelligence (CTI) artifacts such as platforms, vulnerabilities or even natural hazards" in CPS. A CPS security enumerations search engine allows to retrieve information about vulnerabilities related to a search query, for example, the hardware or software product name. In our approach, we compute the cyber risk score for separate CPS components and the entire CPS configuration, including software and hardware, based on registered CVEs. Furthermore, our approach allows automatic software and network reconfigurations in order to replace vulnerable CPS components and harden the CPS.

Jacobs et al. [13] proposed a machine learning-based solution to find best practices in fixing vulnerabilities. Their solution further illustrates the EPSS model [14]. In this model they explore on which vulnerabilities should be fixed first rather than lesser ones that get exploited much less. Unlike our solution which handles processing the cyber risk score for the entire system, their solution focuses on each individual CVE for a particular software or hardware.

Nikoloudakis et al. [15] proposed a machine learning-based solution for an Intrusion Detection System using a Software Defined Networking (SDN). Their approach "detects existing and newly introduced network-enabled entities, utilizing the real-time awareness feature provided by the SDN paradigm, assesses them against known vulnerabilities, and assigns them to a connectivity-appropriate network slice" [15]. While this work is shown to work with Identify, Protect, and Detect vulnerabilities, they are for the individual entities, while our solution is for an entire system of entities or components. Our approach allows to evaluate and reconfigure the existing CPS in automatic mode, based on the CVE database of cyber vulnerabilities.

Adrian R. Chavez [16] implemented moving target defense by randomizing IP addresses and ports of each entity in a control system network with nine entities. Their strategy changed the IP addresses and port numbers in a range of frequencies varying from one second to setting static addresses. They used switches capable of Software Defined Networking to manage the moving target defense implementation. A device they called the randomization controller was added to the network to communicate the new addresses

and ports to all the devices whenever a change was made. An additional router was added to the implementation to handle out-of-band communications between the switch and randomization controller. In addition to the moving target defense, the hosts in the network are running machine learning algorithms to detect anomalous behavior in the network. In contrast, our CMTD solution can be implemented into most environments without required additional hardware.

Sailik Sengupta [17] discussed using AI algorithms to help make MTD more efficient and how to use MTD to make AI agents more secure. To make MTD more efficient, they used a repeated Bayesian game to determine the best switching scheme for the agents in a web application scenario. To help the algorithm with figuring out if a move is good or bad, they mine data from the CVE database and use the scores from CVSS to generate reward values for the AI algorithm. Our VERCASM-CPS engine triggers CMTD based on cyber risk scores of CPS components in order to replace or reconfigure vulnerable components. Our CMTD switching scheme will also be modified by the cyber risk evaluation results. For example, more vulnerable CPS configurations would require more frequent configuration changes.

Markakis et al. [18] proposed a system to protect healthcare networks from devices that are vulnerable to common cyber threats. The system continuously monitors devices that are connected to the network, and new devices before they can connect. Devices are assigned to different virtual LANs, where they have differing levels of access based on the safety score of the device. In contrast, our solution provides vulnerability analysis for each individual software and hardware component, as well as for the entire system. If the software is unsafe, our Controlled Moving Target Defense process will change the software to a different version or replace it with a different software. Our solution also hardens the system by changing its configuration, for example, IP addresses.

Markakis et al. [19] discuss the use of an accelerated device in the network edge to increase computational power and storage of edge devices. Our solution relies on the information about software and hardware versions on each device being shared with the VERCASM-CPS engine. If VERCASM-CPS is implemented in an environment using computationally weak sensors and actuators, accelerators may be added between the CPS hardware and the VERCASM-CPS engine to provide the computational power needed to communicate this data, and to continue its normal tasks.

Toupas et al. [20] proposed a deep learning model for an IDS. This approach can be integrated with our solution, which can spot vulnerable software and hardware components in the system. The most vulnerable components are more likely to be used by network intruders and be a root cause of an anomaly.

## 3. VERCASM-CPS Core Design

In designing our VERCASM-CPS solution, we used a microservice style architecture. This leverages the efficiency of resources and allows VERCASM-CPS to more broadly serve clients in CPS. Monoliths require an overcompensation of services, which are barely used, or parts of a system that do not need the resilience of more expensive hardware. With a microservice design, we fully support a REpresentational State Transfer (REST) implementation [21]. As seen in Figure 1, we have three main microservices: Application Programmable Interface (API) Gateway, Discovery Service, and an Analytics Service. Each have their own role defined in the system. Each microservice uses a RESTful process in which they communicate with each other. Each of the microservies besides the API gateway is using Spring Boot for Java Development Kit (JDK), version 11. Spring Boot allows VERCASM-CPS to easily adapt to our needs with use of dependency injections. A simpler representation of data flow can be seen in Figure 2. We will explain below how each service works and why it is unique to a monolithic design and also how the data flow through our system.

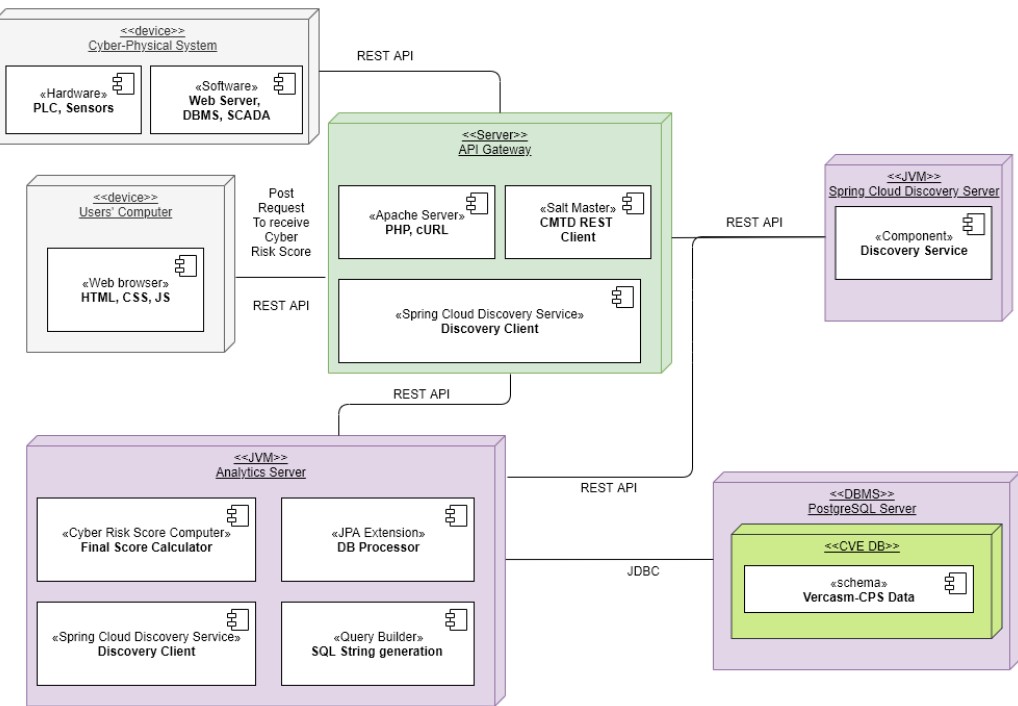

**Figure 1.** UML Diagram of VERCASM-CPS.

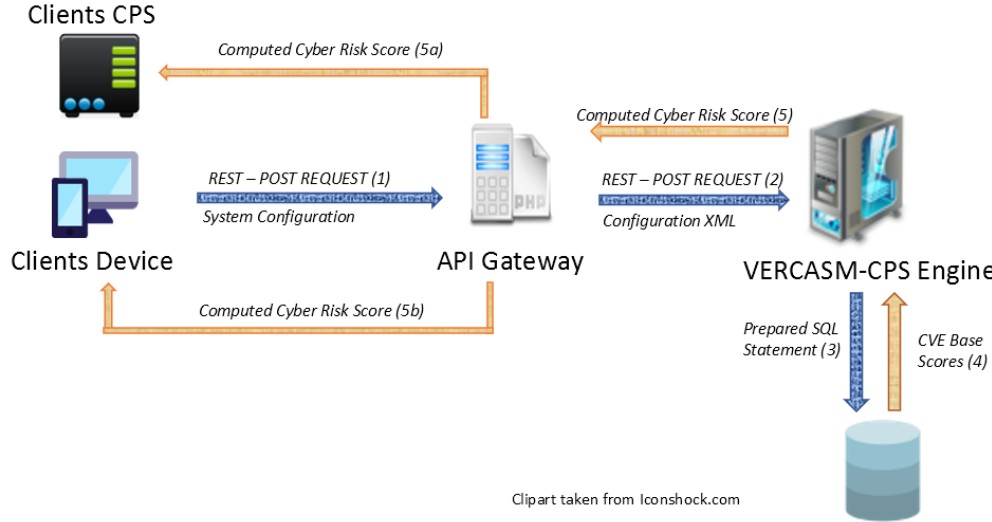

**Figure 2.** General Workflow of VERCASM-CPS.

*3.1. Discovery Service*

We must first discuss the role of our Discovery Service (DS). This allows each service to talk with each other without hard coding ports or IP addresses, as these types of data can change on-the-fly. An incentive for using a service discovery is the fact that it is easier to load balance when a system administrator needs to load more services due to high traffic times. For example, around the holidays major retailers may need to add more services on their web servers to handle more database retrievals. If it were to be hard coded, each service would need to be gone through and manually changed to represent each new end point. This is time consuming and is not efficient in a dynamic environment that servers require.

In making a service discovery, there are two main schools of thought, client versus server side designs. In client side discovery, the client is not the actual user, but instead

another service. A client sends an initial request to a service registry to find the service it wants to talk to. In our design, the client or the API gateway, sends a request to the discovery server and wants to know how it can communicate with the analytics service. This model is familiar with most individuals as we use this in our daily lives. For example, a user wants to shop online. A user may need to find the website on Google to go there. Google is the service registry, the other website is the service a user needs, and a user is the client. One disadvantage however is that the client must make two calls: one to the service registry and the other to the service it needs. On the other hand, there is server side discovery. The client sends one request, and the DS forwards that request to the service it needs. A real world example of this is a large call center. If the client calls one main phone line, a receptionist will forward the client's call to the right department or person, rather than giving the client another number to call back with. One disadvantage with this design is the added complexity.

With our design, we use a client side discovery model. Our design runs Eureka [22]. Eureka was originally developed by Netflix® and made open source with collaboration of Spring Cloud®. With Eureka®, it handles all the necessary functions in which a client or service can look up the name of a service then send the requests. One major benefit of Eureka that incentivized our choice was the advanced caching it has. If a service route disappears, it caches the previous route for that service so the client will not have to wait for the connection to re-appear. We must also discuss the discovery clients in the different services and gateways. It sends out heartbeats to the service registry as a service. These are sent in case another service needs to know where it is located on a server or in the cloud. This allows the service to be dynamically load balanced.

### 3.2. API Gateway

The API Gateway is responsible for handling forward facing requests. Forward facing means that if the user would execute a Domain Name Service trace to the front end, they would only see the IP address of the Gateway. Its main job is to act like a router. Thus, depending on what the user wants to see, it will route traffic to the services that are allowed for that user. In the case of our design, we will introduce an example and discuss each subsequent service, show how it interacts with each service, and why that service is needed. In our example, we first initialize the request from the front end for an Operating System (OS) and a SCADA system through a RESTful POST request. A generated CPS configuration structure is then drafted in the API Gateway to be sent. This is illustrated in Listing 1, using pseudocode. The CPS configuration may include an operating system: Debian [23] version 10, type 64 bit, and one program: the SCADA system "Ignition"® version 8.0.17 [24]. The API Gateway is written in PHP [25] (This solution is not endorsed or made by The PHP Group) language, rather than our back-end system, written in Java, later discussed. Using RESTful API and microservices, we can implement algorithms in different languages that do certain tasks the best. The main computation occurs in a Java Virtual Machine (JVM) [26]. A front facing website running purely in a compile type design makes the developer to stop the server and re-compile every time a change is needed. With PHP being a scripted language, we can change code on-the-fly, such as HTML that is served to the client. This leaves the computationally intensive services able to run on more dedicated machines. Having a web interface allows us to implement a "What If" functionality over a software scanner that allows users to test their entire system with certain CPS components rather than being forced to only evaluate what they already have in terms of installed software or hardware.

**Listing 1.** Generated CPS Configuration Structure (in pseudocode) sent from API Gateway to Analytics Service (VERCASM-CPS Engine).

```
Operating System Name: OS-name
Operating System Version: OS-version
Number of programs: N
Program 1 ID: id 1
Program 1 Name: program-name 1
Program 1 Version: program-version 1
...
Program N ID: id N
Program N Name: program-name N
Program N Version: program-version N
```

### 3.3. Analytics Service

We will continue our example here in the Analytics Service. Once a POST request is received from the API gateway, it stores the necessary information to query the PostgreSQL® Relational Database Management System (RDBMS) [27]. This service ultimately will return the cyber risk score back to the API Gateway then back to the client's web browser. We will discuss each sub-component of this service, in the likeness of a method or function that this service provides: Query Building with Database Connections, the Cyber Risk Computer, and the CMTD.

#### 3.3.1. Query Builder

The data arriving from the API Gateway are stored in the Java objects. These objects hold the retrieved data from the generated CPS configuration structure, as seen in Listing 1. The data are then sent to a query builder where each query is built upon a base template. First, we must discuss how Common Platform Enumerations (CPE) are composed. Each CPE follows a simple template in which a colon, ':', is used as a delimiter as seen in Listing 2. For each "part" it has three possibilities as shown in Table 1.

**Listing 2.** CPE Definition.

```
{part}:{vendor}:{product}:{version}:{update}:{edition}:{language}
```

**Table 1.** Part Possibilities.

| Notation | Definition | Example |
|----------|------------|---------|
| a | Application | "Ignition"® |
| o | Operating System | Red Hat® Enterprise Linux ® [28] |
| h | Hardware | Modicon® M221 |

In our example, we use Debian® as the OS which will be denoted 'o'. In some CPEs, the version which that application or OS refers to is not in the CPE but rather to other keys, such as 'version start including' or 'version end including'. Hence, we construct Regular Expressions (REGEX) in the query builder. In Listing 3, line 17 and line 34 are for matching CPEs that have the name of the software and the version included. The second REGEX is where there is no version included in the individual CPE as shown in Listing 3, line 23. This REGEX is queried twice, once for if the CPE contains no version as mentioned and the other for if the versions are in other columns. We designed our system so that if a CPE contains no version and there are no versions in other keys, we assume that this CVE is affected by all versions of that software. Note that, in the REGEX inside of our SQL statement of Listing 3, 'PART' is replaced with an *o*, defining the Debain OS.

**Listing 3.** SQL Query of an OS.- confirmed.

```sql
SELECT
DISTINCT cve_name,
CASE
WHEN version3_base=NULL then
version3_base=version2_base
WHEN version2_base=NULL then
version2_base=5.0
END,
COALESCE(version3_base,5.0) as version3_base,
cpe_column
FROM vercasm-cps_data
WHERE
(
cpe_column LIKE '%debian%10%x64%'
AND
cpe_column  ~'cpe:2\.3:[o]:([\w]{0,40}):([\w]{0,40}):([a-z\d]{0,40})'
)
OR
(
cpe_column LIKE '%debian:%'
AND
cpe_column  ~ 'cpe:2\.3:[o]:([\w]{0,40}):([\w]{0,40}):\*:\*:\*:\*:\*:\*:\*:\*'
)
OR
(
cpe_column LIKE '%debian%'
AND
(
version_start_including LIKE '%10%'
OR
version_end_including LIKE '%10%'
)
AND cpe_column ~ 'cpe:2\.3:[o]:([\w]{0,40}):([\w]{0,40}):\*:\*:\*:\*:\*:\*:\*:\*'
)
```

We begin our SQL query to SELECT each 'cve_name' attribute, which is what VERCASM-CPS uses as an ID. This is what a client commonly references to the name of a CVE, for example, *CVE-2021-33910*. Since multiple CPEs can be determined for each CVE, we use *DISTINCT*. This only returns the base score of each CVE for any CPE that matches with it. Each CVE has its own CVSS. Each CVSS is based on a wide range of metrics such as attack vector, user interaction, and the Confidentiality, Integrity, and Availabilty impacts [5]. This comes in two referenced versions CVSS2 and CVSS3, the latter being newer. We designed our query to prioritize the newest version of the scoring system. However, some CVEs, which can date back to 1999, only contain a version 2 score. Thus, we developed a case system that follows several rules as seen in Table 2.

**Table 2.** Breakdown of Case Options and Coalesce in Listing 3, lines 3–12.

| If Statement | Then Statement |
|---|---|
| If CVSS3 is null | Then Automatically assign it to the CVSS2 score |
| If CVSS2 is null: | Then Set it to a median score of 5.0 |
| If any score remains as NULL | Set it to the median score of 5.0 |

On occasion, the CVSS version 2 or version 3 contains no base risk score; this can have an affect on our system. The other is the occasion where a cyber risk score was never assigned for a CVE. In both cases, the CVE cyber risk scores have yet to be assigned. By default, we assign the CVE cyber risk score as 5.0, which is the median value for the interval from 0 to 10 with step 1. For new software and hardware releases, the CVEs are not discovered and published for some time after their release. In this case, the CVE represents a zero-day vulnerability, which is unknown. We assume that the newly released software or hardware is not perfectly secure and their vulnerabilities will eventually be discovered and published in the NVD. We assign the cyber risk score as 5.0 for the new software or hardware components. We think it is more practical than conservatively assigning a critical score 10.0 or a score 0 which indicates that the CPS component is perfectly secure.

To retrieve the CVE records for a given software or hardware CPE that exists, the SQL query shown on line 17 continues to check the cpe_column attribute. This represents the individual CPEs. We check three possibilities of CPE enumerations. First, we check if the CPE contains the name of the software or hardware, version, and type. Note that the software or hardware name must be before the version, and the version must be before the type. In our example Debian 1064 bit , it must also match our REGEX shown on line 17 denoted by *AND* keyword. Second, we check if the CPE only contains the name of the OS. It is common for Microsoft® OS to use the notation of Windows® 10: *windows_10*. To check in that example if it affected all Windows®, and not the *_10* part, we put the OS next to the delimiter. It must also match the REGEX on line 23. Finally, as mentioned above, we check other columns in which the CPE does not contain the version numbers but are in other keys. To check these two, we use the *OR* keyword. However, in order to match, it must match line 34 as well. For the software, we can see the *PART* is reference by an *o*, and further applications would be denoted by an *a*. Each subsequent query is a *UNION* each of the previous queries. This accomplishes a few items: reduces the number of open connections and minimizes planning time taken up by the RDBMS.

### 3.3.2. Java Persistence API Repository

Once the query is formatted and validated, we use an extension of Java Database Connectivity (JDBC) named the Java Persistence API, or JPA, which is an extension of paging and sorting, to an extension of CRUD: Create, Read, Update, and Delete. More specifically, extensions are Java® packages that extend the core functionality of previously defined Java® classes. The JPA extension, as a repository, supplies basic built-in queries to do simple queries such as selecting IDs or updating tables. For our complex query, we use a JDBC template that is sent over to the Postgres® RDBMS. JPA handles transactions and normal RDBMS operations in which a developer may need to handle manually. An example of this can be seen in Figure 3.

### 3.3.3. Cyber Risk Computer

When considering how each of the vulnerabilities for a piece of CPS software or hardware should be represented, it was decided that we will rely on weighted averages to compute the total CPS cyber risk score. Weighted average would allow for the vulnerabilities with higher CVSS scores to heavily influence the mitigation tactics taken. This choice also allowed for software with few but high-rated scores to be taken into account and not simply cast aside. Lastly, because the number of vulnerabilities is finite, a weighted average suited the data. The mathematical definition of a weighted average follows.

**Definition 1** ((Weighted Average) [29]). *$x = \frac{\sum_{i=1}^{n} w_i x_i}{\sum_{i=1}^{n} w_i}$, where each weight $w_i$ is a non-negative real number and each $x_i$ is a data point and n is the number of said points.*

The ranges of the CVSS scores given by NIST were taken into account. Instead of simply assigning each data point a different weight, each range was assigned a weight and each number inside that range had that weight applied. This allowed for the average to be influenced by these weights and tend toward the necessary range. To obtain the necessary

gaps, the weights to be assigned had to have enough differentiation to create them. Thus, different set weights needed to be evaluated to find the lone set that allows for the weighted average to land in the range needed. Table 3 depicts the three set of weights evaluated.

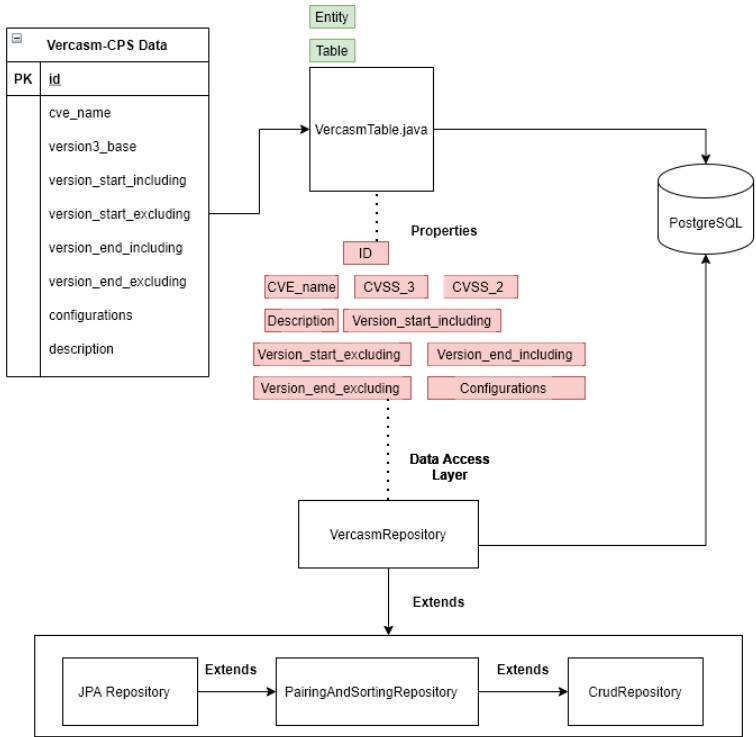

**Figure 3.** VERCASM-CPS Database Workflow.

**Table 3.** Weights assigned per range.

| Severity Ranges | Weights A | Weights B | Weights C |
|:---:|:---:|:---:|:---:|
| Low (0.1–3.9) | 0.1 | 0.1 | 0.1 |
| Medium (4.0–6.9) | 0.2 | 0.2 | 0.2 |
| High (7.0–8.9) | 0.5 | 0.6 | 0.9 |
| Critical (9.0–10.0) | 1.5 | 1.6 | 4.4 |

These weights were developed with interval creation in mind, as well as the Fibonacci sequence, a sequence used frequently in computer science.

Weights A were computed using the following formula:

$$w_i = w_{i-1} * t_i, \tag{1}$$

where $t_i \in [2, 2.5, 3, 3.5...]$.

Weights B were computed with the following formula:

$$w_i = (w_{i-2} + w_{i-1}) * 2. \tag{2}$$

Weights C were computed using the formula:

$$w_i = (w_{i-2} + w_{i-1}) * t_i \tag{3}$$

where $t_i \in [2, 3, 4, 5...]$.

The range in which the weighted average tends needed to fall into one of the predefined ranges given by NIST (Table 3). Thus, it needed be shown that the range does not

tend toward a negative number and that it does not exceed 10.0. The weighted average with the weights defined in Table 3 will not tend toward a negative number due to both sums involved are sums of positive numbers. This is due to the fact that the weights and numbers in the ranges are positive, and hence, the multiples and subsequent sums will be positive. It is left to show that the average in question will not exceed the boundary of the ranges (10.0).

**Theorem 1.** *Let $x = \frac{\sum_{i=1}^{n} w(x_i)x_i}{\sum_{i=1}^{n} w(x_i)}$ where each weight is defined by the piece-wise function:*

$$w(x_i) = \begin{cases} 0.1, & x \in [0.1 - 3.9] \\ 0.2, & x \in [4.0 - 6.9] \\ 0.9, & x \in [7.0 - 8.9] \\ 4.4, & x \in [9.0 - 10.0] \end{cases},$$

*each $x_i \in [0.1 - 10]$ and n is the number of CPS components evaluated. If $x, x_i$, and $w(x_i)$ is as defined, then $x \leq 10.0$.*

**Proof.** Let $x, x_i, w(x_i)$, and n be defined as in the above Theorem 1. To show that $x \leq 10.0$, we must show that a finite number of the highest possible value for $x_i$ with the highest weight $w(x_i)$ is less than or equal to 10.0. Hence, we let $x_i = 10.0$ and $w(x_i) = 4.4$. Therefore, $x = \frac{\sum_{i=1}^{n} w(x_i)x_i}{\sum_{i=1}^{n} w(x_i)} = \frac{\sum_{i=1}^{n}(4.4)(10.0)}{\sum_{i=1}^{n} 4.4} = \frac{n(4.4)(10.0)}{n(4.4)} = 10.0$. Thus $x \leq 10.0$, as desired, and since all other possible combinations of weights and possible values of $x_i's$ are less, then the weighted average stays in the necessary range. □

In continuation of our example, if Linux Debian OS returns 11,360 CVSS scores and "Ignition"® SCADA returns 21 CVSS Scores, they are calculated in our score computer equally. We consider each vulnerability from every component, in our design. For example, if a vulnerability comes from the OS or SCADA we treat them as if either will contribute to increase the attack surface. As seen in the query builder, each software and hardware component, including the OS, is considered equally adverse to vulnerabilities in a system. In our example, cyber risk scores are calculated using the same weights, in a pool of 11,360 + 21 = 11,381 CVSS scores, to compute the total CPS cyber risk score for an entire system.

*3.4. CVE Database*

Each CVE year can be downloaded from the NVD. Each subset of CVE years is in the form of a Javascript Object Notation (JSON), which then is further processed into our RDBMS, Postgres®. With the JSON searching, we pull out the CVE id, CVE description, CPE, base score, and any version-related content represented for each CVE.

In planning our model, we consider every vulnerability in each CPS component not independently to each other. System administrators and CPS operators input all the software on their devices and the hardware in use into our VERCASM-CPS framework. We search all related CVEs and CPEs for the keywords of all hardware, software and the operating system given. We use a future focused model that shows the worst case scenario for software installed on a user's computers. We also provide a "What If" functionality. This functionality allows the user to track the CPS cyber risk score either over time or through experimentation in designing a system. VERCASM-CPS allows for system administrators to know collectively in advance whether to install one program or another, one piece of hardware or another. This allows them to see how vulnerable their CPS is or could be. Other models of similar type to ours seem to only consider certain types of attacks or prefer to see the source code. While these models are beneficial in their own right, VERCASM-CPS disregards the source code and the type of attack allowing all types of both to be considered on an even level. VERCASM-CPS is system-agnostic for the cyber risk analysis and can run wherever a JVM may execute.

### 3.5. Controlled Moving Target Defense

Once the cyber risk score has been evaluated and sent back to the user, if it is above a specified threshold set by system administrators, CPS operators, and engineers, our CMTD solution ensures that the software configuration of the CPS is always the most secure based on the known vulnerabilities gathered from the CVE database. It also hardens the CPS by changing IP addresses and port numbers of the devices in the CPS network. The software configuration changes are handled by the automation and infrastructure management software Salt [30]. The network configuration changes are performed by scripts located on CPS devices that have enough computational power. Once a significantly large cyber risk score has been detected, Salt will be activated and the scripts on the device will be executed. These scripts are written in shell script. Salt can also change installed software components on the device. Thus, if for example, a web browser has a high cyber risk score determined by the VERCASM-CPS engine, Salt can uninstall that web browser and install a more secure replacement. The workflow of VERCASM-CPS can be seen in Figure 4.

Salt can be run on any computer that can install the Salt software and connect to the local network. It is based on a master and minion system. This is where the minions know the IP address of the master, and the master can send instructions to the minions. Instructions can be sent to any number of minions at once, so it is possible to change the configurations of every device at once or individual devices. In order for Salt to work, the master and minion process must be running on the respective computers. The master process should be running on the same computer that detects CVE scores, so there is no network communication required to issue commands to the minions. It is important to note that a Salt master can also be a minion, so if the computer detects a high CVE score on itself, Salt can handle the configuration changes and make it more secure.

A Salt master can also check current hardware usage for several components of its minions. By using the status module, a Salt master can read information about the current usage of the CPU, memory, and storage of a minion. This is a useful tool that allows for remote monitoring of the minions, which may lead to the detection of malware that is running on the minion. Once anomalous behavior has been detected, the Salt master can use tools from the 'ps' Salt module to see the running processes of a minion and kill any that are suspect. It can also see the usage of each resource that a process is taking, for example how much memory or CPU usage a process is using, which can also help find any programs that are taxing on the performance of the machine, or have crashed and are consuming resources.

CMTD is a process that applies to both the CPS software and hardware. If we find that there is no safe configuration with Linux Ubuntu, we can go in and change the operating system to Microsoft® Windows® 10 with a more secure build. Once the operating system is changed, the Salt minion service will need to be installed and configured again, but once it is running, the Salt master can handle installing packages and software needed by the users.

Our framework can generate recommendations to replace vulnerable hardware in a manual mode. While the replacement is in progress, CMTD can reconfigure vulnerable hardware to make it more resilient against cyber attacks. Since we do not control the software itself, like in the EPSS model for developers, we use CMTD to move configurations around to not arrive at the same vulnerabilities. For example, CMTD can change the hardware network address. Examples of hardware vulnerabilities include the Meltdown and Spectre vulnerabilities that affect most modern CPUs, including those manufactured by AMD, Intel® (Intel is a trademark of Intel Corporation or its subsidiaries [31]), and ARM™ (AMBA, Arm, Arm ServerReady, Arm SystemReady, Arm7, Arm7TDMI, Arm9, Arm11, ... are trademarks or registered trademarks of Arm Limited (or its subsidiaries) in the US and/or elsewhere. The related technology may be protected by any or all of patents, copyrights, designs and trade secrets. All rights reserved [32]).

To summarize, CMTD can be used to change the following things in the system:

- IP Addresses
- Installed software
- Operating System
- Hardware
- Memory layout with address space layout randomization
- Sensors and peripheral devices connected to the system

**CPS Configuration
(Hardware and Software)**

**Figure 4.** Integration between VERCASM-CPS Engine and Controlled Moving Target Defense.

## 4. Evaluation

In testing our core design of VERCASM-CPS, we developed a Cartesian product model where we tested for the mean, geometric mean, and the three classifications of a weighted average. The basis of a Cartesian product allows us an insight into the idea that the size of the categories need not be the same in order to analyze them and how many types of software and hardware configuration setups are possible. This is important to our work because, as previously mentioned, we want our users to be able to design their hardware and software configuration in advance by taking into consideration the most secure software and hardware options. The definition of a Cartesian product follows.

**Definition 2** ((Cartesian Product) [33]). *If A and B are sets, the Cartesian product of A and B is the set:* $A \times B = \{(a, b) : (a \in A) \, and \, (b \in B)\}$.

There is an important property of Cartesian products: "if A and B are finite sets, then $|AB| = |A||B|$ because there are $|A|$ choices for the first component of each ordered pair and, for each of these, $|B|$ choices for the second component of the ordered pair [33]." Since the contents of the categories are finite in our case, we can apply this property to find the number of possible configurations when we examine different sizes of these categories. Thus, since we have six categories of four components being evaluated in our design, this implies $4 \times 4 \times 4 \times 4 \times 4 \times 4 = 4^6 = 4096$. If we add another two versions of software to any of the categories then it becomes $6 \times 4 \times 4 \times 4 \times 4 \times 4 = 6 \times 4^5 = 6144$. This will also allow users to anticipate how many different ways they could possible configure their hardware and software system.

In our methodology, we followed a few assumptions: Windows® and Linux®.

- Data Gathered from CVE, such as the CVSS is a trusted, reputable source.
- We assume our base hardware and software, such as JVM is trusted.

Furthermore in methodology, we tested using two operating systems (OS) bases, Microsoft® Each base then included two version each as shown in Table 4. We then tested each OS for an inclusion of different software categories. These categories included are shown in Table 5. Categories were chosen based on what an engineer may use on an example of an industrial control system. Each system was tested for all permutations of possible versions and vendors for each OS. For this example, it also aligns with a permutation with replacement model to the same outcome, for which there are four softwares in each category where each version, no matter the vendor, is independent of each other. There are six total categories including one for the operating system. We tested $4^6$, 4096, permutations. The system was tested in a sequential format, where the the first permutation ran first up to the number of permutations. However, after choosing our classification, we also tested Linux Ubuntu, adding another 2048 iterations. Below is our system configuration for conducting tests with VERCASM-CPS including the RDBMS, which was hosted on the same computer.

- CPU: AMD® Ryzen® 7 3700X @ 4.4 Ghz; RAM 32 GB DDR4 @ 3200 Mhz [34].
- OS: Microsoft® Windows® 10 Pro, 64 Bit
- Web Server: Apache®, Apache® Tomcat® with Spring Boot® [35]
- JDK: AdoptOpenJDK Version 11

**Table 4.** Operating Systems Tested.

| Name | Version |
|---|---|
| Red Hat Linux® | 8.3 |
| | 8.2 |
| Windows® 10 | 20H2 |
| | 2004 |
| Ubuntu® | 20.04 |
| | 18.04 |

**Table 5.** Software Tested.

| Name | Version |
|---|---|
| Web Browser | |
| Firefox® | 88 |
| Firefox® | 89 |
| Chrome® | 86 |
| Chrome® | 89 |
| Database Management System | |
| Oracle RDBMS® | 18c |
| Oracle RDBMS® | 19c |
| PostgreSQL® | 12 |
| PostgreSQL® | 13 |
| Web Server | |
| Nginx ® | 1.17.10 |
| Nginx ® | 1.19.7 |
| Tomcat® | 9.0 |
| Tomcat® | 10.0 |

**Table 5.** *Cont.*

| Name | Version |
|---|---|
| Achiever | |
| Peazip ® | 6 |
| Peazip® | 7 |
| 7zip® | 19 |
| 7zip ® | 20 |
| SCADA | |
| "Ignition"® | 8.0.17 |
| "Ignition"® | 7.9.17 |
| Webaccess/SCADA | 8.3 |
| Webaccess/SCADA | 9.0 |

In choosing the classification of weights, we ultimately chose classification *C*, as seen in Table 3. We tested all three classifications as seen in Figure 5. In this test, we used 4096 permutations of software. We also tested for mean and geometric mean as seen in Figure 6. Our Cyber Risk Score is less conservative, and it better reflects the probability of a given cyber vulnerability to be exploited. Versus other products in this space, our solution works with a wide range of CVEs. We made an effort to ensure that higher CVSS scores are weighted as such. Even though a product may have more vulnerabilities, they all may be minor versus a hardware or software package with few CVSS scores but all above the critical range.

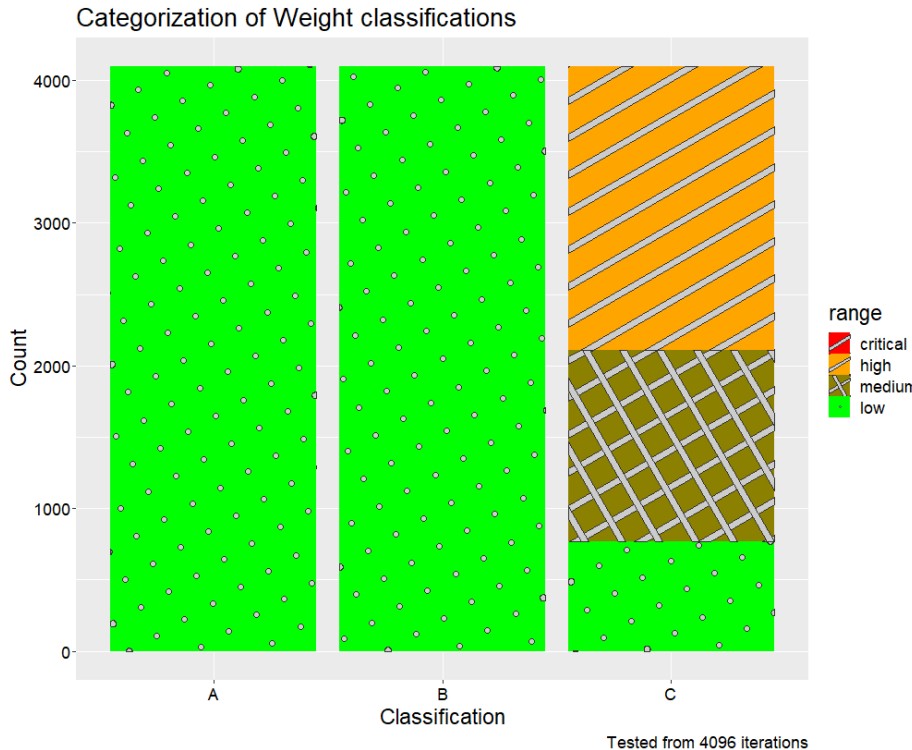

**Figure 5.** Tested Weight Classifications.

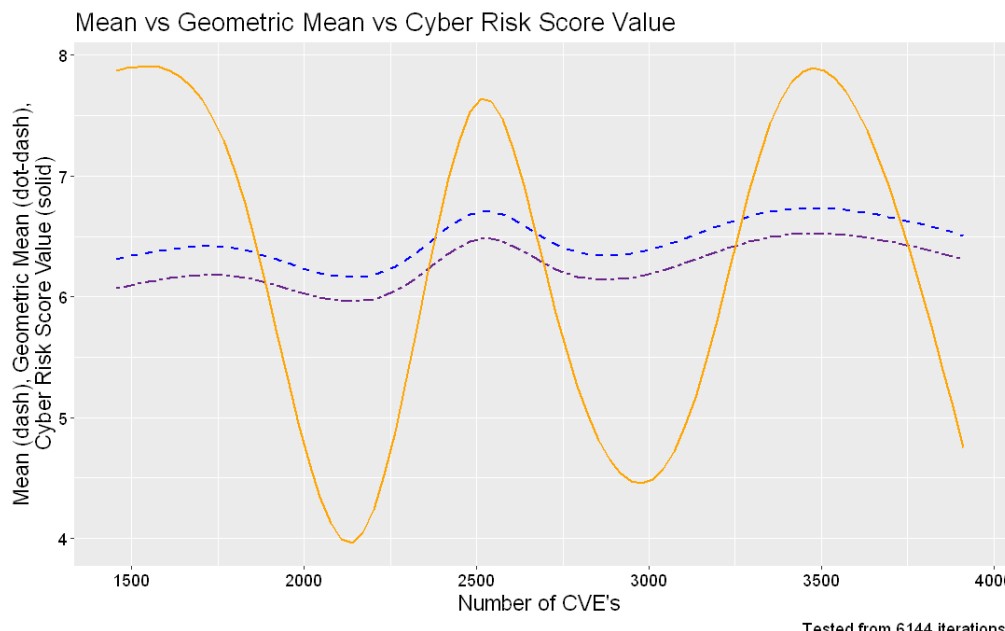

**Figure 6.** Mean vs. Geometric mean vs. Cyber Risk Score.

In one example shown in Table 6, we found many instances where changing the OS or SCADA system to a previous version actually decreased the cyber risk score. In this example, all systems were tested for using Chrome® 89, Tomcat® 10.0, 7zip® 20, for a Web browser, Web Server, and Archiver respectively, the only category which changed was the SCADA version of Webaccess/SCADA between versions 8.3 and 9.0 [26,35–38]. This example shows the benefit of our system of engineer's trying to find the most secure operating system and SCADA system possible. In some environments, Windows® is preferred by the user. In this example, an older build of Microsoft® Windows®: 2004, would be more secure with the newest version of the SCADA. However, if an engineer needed to use an older version of SCADA, it is recommended to use the new version of Windows® 20H2. However, if Red Hat® Enterprise Linux® could be used, it is applicable to use whatever version SCADA of the two which gave the most benefit, being a clear winner in terms of having a lower cyber risk score. With the inclusion of Ubuntu in testing, we found that it had the highest cyber risk score. The stereotypical notion that Linux® is more secure is disproved from Ubuntu's scoring. Just because it is Linux does not automatically make it less vulnerable. The more factors examined, in this case the type of software it is running, the better a result an engineer would receive.

**Table 6.** Subset of cyber risk scores.

| Operating System | Cyber Risk Score |
|---|---|
| SCADA: Webaccess/SCADA 8.3 | |
| Windows 20H2 | 4.01 |
| Windows 2004 | 4.51 |
| RHEL 8.2 | 3.66 |
| RHEL 8.3 | 3.66 |
| Ubuntu 18.04 | 6.43 |
| Ubuntu 20.04 | 6.13 |

**Table 6.** *Cont.*

| Operating System | Cyber Risk Score |
|---|---|
| SCADA: Webaccess/SCADA 9.0 | |
| Windows 20H2 | 6.07 |
| Windows 2004 | 4.51 |
| RHEL 8.2 | 3.66 |
| RHEL 8.3 | 3.66 |
| Ubuntu 18.04 | 6.43 |
| Ubuntu 20.04 | 6.13 |

*4.1. Scalability*

In the interest of scalability, as mentioned before, we chose PostgreSQL® as an RDBMS system for handling relational data processing. This was chosen due to that we are not dealing with what may be seen as big data. Our data are only in the gigabyte range, and we felt that considering the dataset analyzed to be big data would be in error. If our data were more sizeable, we would have chosen to use other systems, such as Apache's Hadoop® framework or Spark®. Fortunately, having the power of a relational database such as PostgreSQL® allows us to have greater adaptability for our system to run on a wide range of server hardware and operating systems. It also has a greater reach of engineers and developers that have experience working with Postgres® and programs like it. We do have plans however to switch from unionizing our SQL queries into batches especially for systems that contains thousands of software components that would need to have a cyber risk score given on an hourly or daily basis. In testing, we also discovered that our system ran faster due to advanced internal caching techniques that Postgres® uses. As seen in Figure 7, query times were faster as the number of queries increased. This is furthers the reasoning in our choice of usingan RDBMS such as Postgres®. While it may seem inefficient with little traffic, it does better not worse with lots of traffic polling the server up to a certain point. The microservice architecture also helps this where traffic becomes overwhelmed in the system. It is noted that total query time was taken from the microservice side of the run, where normal network delays and transactional processing done in spring boot's JDBC is expected, but minimal in the measurement of time as seen in Figure 8, where RTT is never than a few milliseconds above the total query time.

*4.2. Controlled Moving Target Defense*

To test the speed of our CMTD mechanism, we created a test bed that includes a powermeter from "Schneider Electric"®, Raspberry Pi, and a laptop running the "Ignition"® SCADA system. The version of "Ignition"® we chose is called the "Maker Edition" [24]. It is a version of "Ignition"® that is licensed for educational or personal use only. Our Raspberry Pi model is the Raspberry Pi 3 Model B. The overall flow of data can be seen in Figure 9. The powermeter reads information about devices connected to it, such as voltage and current, then sends those to the Raspberry Pi via the MODBUS® protocol. The Raspberry Pi reads these values with an OPC UA server that is written in the Python programming language using the FreeOpcUa library [39]. These values are then converted to OPC UA tags and are sent to the OPC UA client built into "Ignition"®. A SCADA client can be connected to "Ignition"® via a web browser on any device. Operating screens can be configured to see these values in a convenient way for the users.

The connection between the OPC UA server running on a device and the OPC UA client running in "Ignition"® is secured by the Basic256Sha256 security policy [40] created by the OPC Foundation. The security mode used for the policy is "Sign and Encrypt". This mode provides data confidentiality and integrity for the communication channel. Confidentiality is provided by encrypting the data with the AES256-CBC algorithm. Integrity is provided because the sender signs the data with their private key, which is generated from the HMAC-SHA2-256 algorithm.

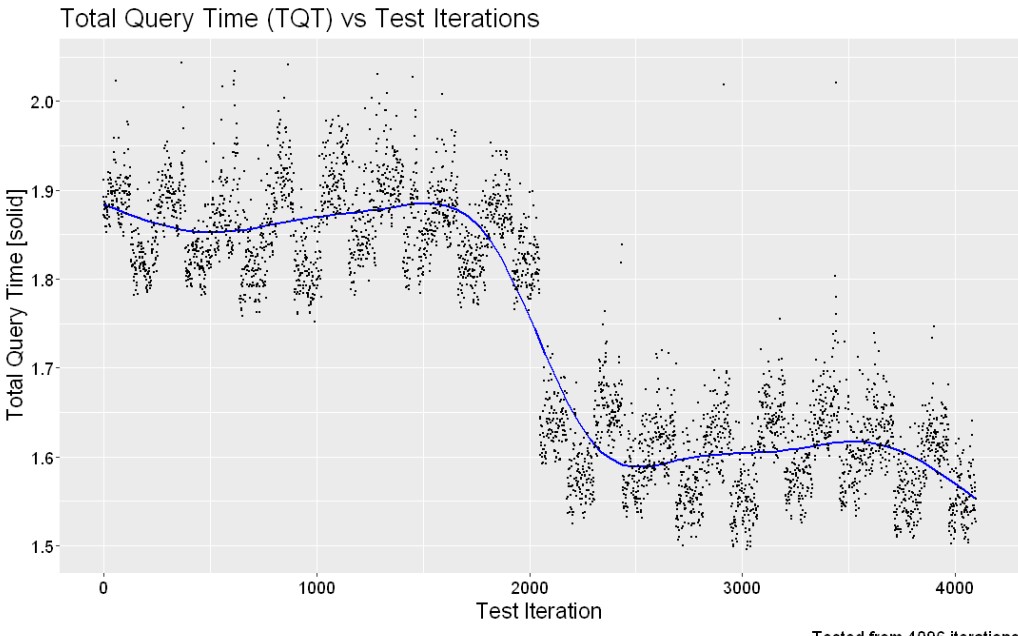

**Figure 7.** SQL Query Time vs. Run Iteration.

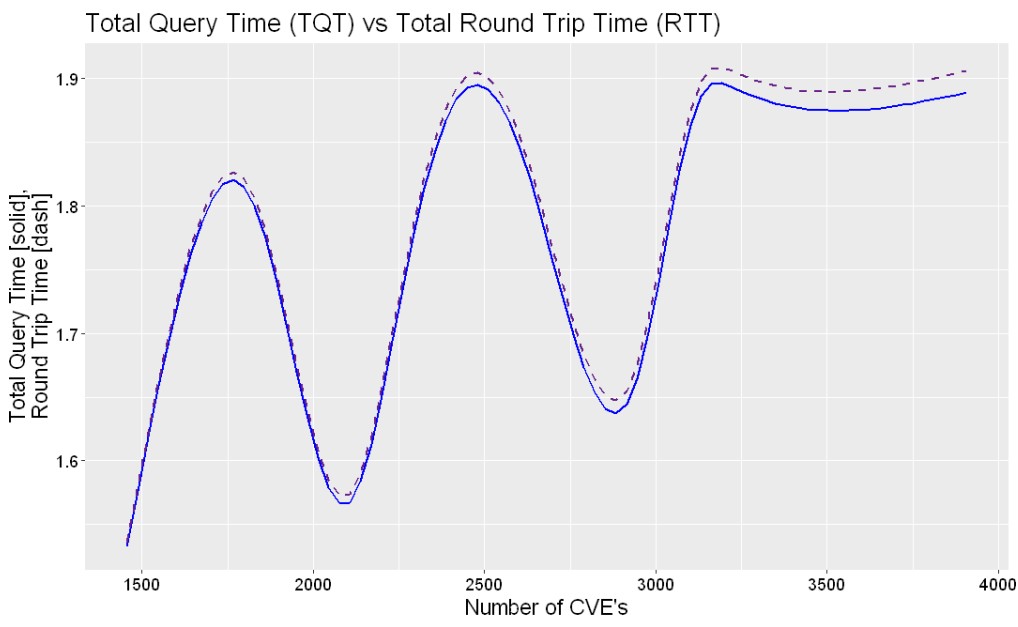

**Figure 8.** SQL Query time vs. Round Trip Time.

For added security, between the OPC UA server running on the Raspberry Pi and the OPC UA client running in "Ignition"®, we generate noise in the form of randomly generated OPC UA tags. The values that can be assigned to the tags follow a discrete uniform distribution. They can be configured to be in any range that the user wants to set, so any value within that range will have equal probability to be used for the fake tags. These fake tags are sent to the "Ignition"® SCADA system through the same channel as the tags that hold the real values read from the power meter. In order to not confuse the operators of the SCADA system, the fake values are not visualized on the screen of the SCADA client. However, since the OPC UA client on "Ignition"® is still receiving the tags through the communication channel from the OPC UA server, if any attackers are theoretically able to break the encryption and see the data in plaintext, they will see both

the real and fake tags, and must determine which are the real tags representing data from the power meter.

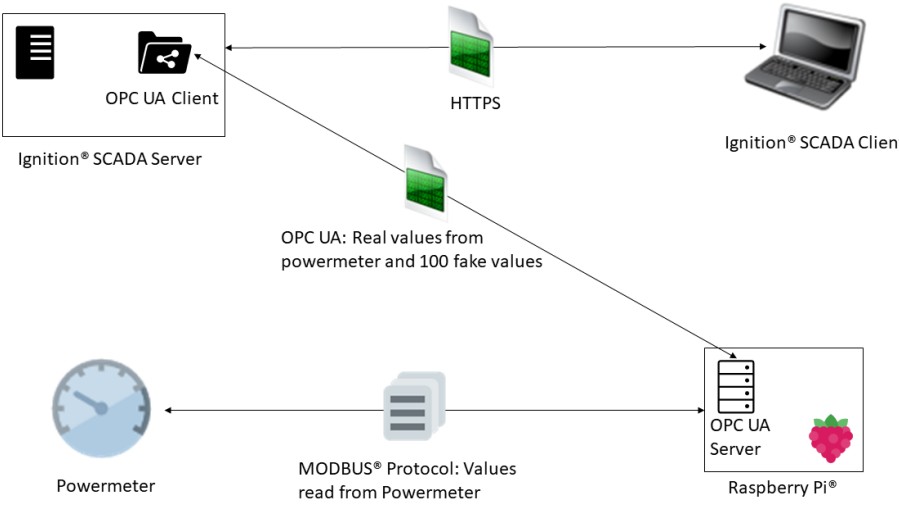

**Figure 9.** MTD Test Architecture.

The key component of this CMTD procedure is changing the IP address of the Raspberry Pi. The scripts that change the network adapter properties of the Pi are Bash [41] shell scripts. The scripts use Linux commands to change the IP address of the device they are located on by changing the IP of the network interface. Once the IP of the network interface has been changed, the script stops the running OPC UA server and starts a different one that is configured to run on the new IP address. The new OPC UA server is also configured to run on a different port than the previous one. Our CPS test bed includes a computer that is running the "Ignition"® Supervisory Control And Data Acquisition (SCADA) server. "Ignition"® has a connection preconfigured for both of the OPC UA servers, so whenever the new server starts, the connection is automatically established and data from the sensors can be transferred again. The number of scripts and OPC UA servers on the device depends on how many different IP addresses the system administrators want to switch to. Below is the description of the system configuration where the experiment was conducted:

- SBC: Raspberry Pi 3® (Raspberry Pi is a trademark of the Raspberry Pi Foundation [42]) Model B
- CPU: Quad Core 1.2 GHz Broadcom® BCM2837; RAM 1 GB DDR3
- OS: Raspberry Pi OS®; Kernel 5.10 64 Bit
- SCADA: Ignition Maker Edition™ version 8.1.1; Ignition Designer™ version 1.1.1

The evaluation of this test bed measures the time it takes from starting the CMTD process to when it finishes. The process stops the running OPC UA server on the Raspberry Pi, changes the IP address of the Raspberry Pi, then starts a new OPC UA server. The measurements start when the running OPC UA server is stopped, and end when the "Ignition"® SCADA server connects to the new OPC UA server. There were 202 measurements taken from this experiment, and the results can be found in Figure 10. We found that the maximum time for "Ignition"® to connect to the new OPC UA server was 11.353 s, the mean was 9.722 s, and the minimum was 9.473 s. A lot of this time can be contributed to the OPC UA server starting, then "Ignition"® polling for the new connection.

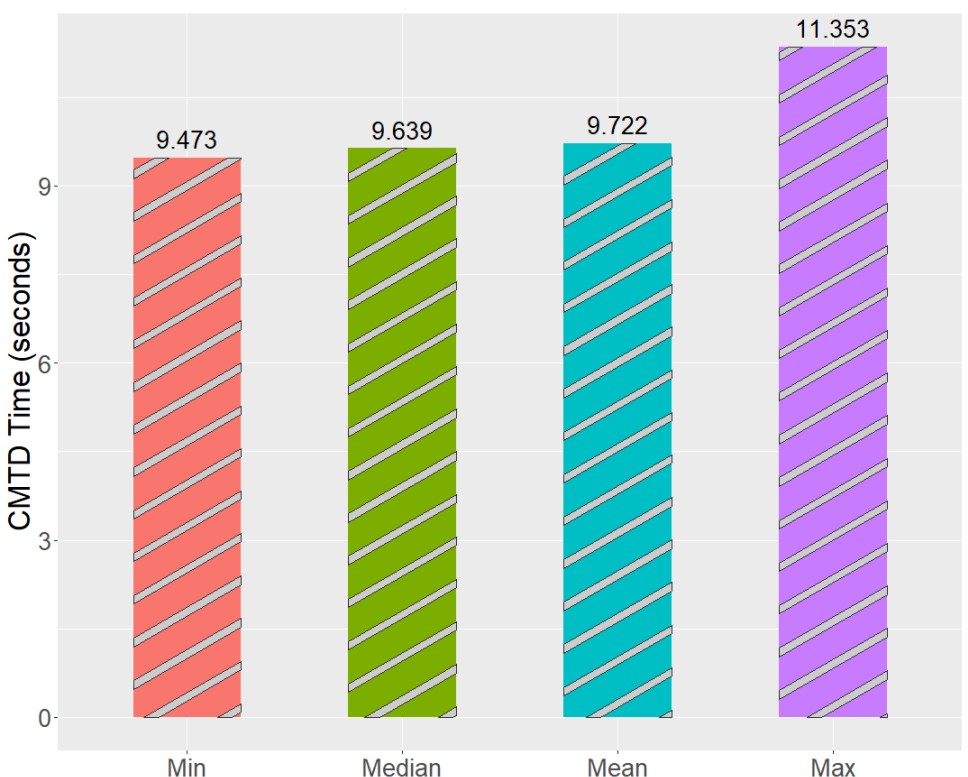

**Figure 10.** MTD Test Results.

## 5. Conclusions

In this paper, we presented a CVE-based methodology to evaluate the cyber risk score of each separate CPS component, including software and hardware components, and the total CPS risk score in automatic continuous mode. Our solution allows CPS designers and operators to determine the most secure CPS configuration for newly designed CPS and components to reconfigure or replace existing CPS. The "What If" functionality allows us to evaluate how replacing one or several CPS components would affect the total CPS cyber risk score.

We conclude by testing our VERCASM-CPS cyber risk evaluation engine that some Linux® distributions are less secure than Microsoft® Windows® 10. VERCASM-CPS showed that even a well developed Linux® distribution such as Canonical® Linux Ubuntu® *20.04* and *18.04* is less secure than Microsoft® Windows® 10 in build numbers *2004* and *20H2*; this is shown as of 31 July 2021. It should also be noted that Red Hat® Enterprise Linux® is more secure than Canonical® Linux Ubuntu® *20.04* and *18.04* , even though they have the same underlying kernel controlling both of them. This illustrates that, while the kernel is important, it is not always the true root cause of cyber vulnerabilities. Evaluating the cyber risk of separate CPS components independent of each other allows us to identify the most vulnerable CPS components that have a higher risk of being exploited.

Our Controlled Moving Target Defense solution is integrated with the VERCASM-CPS cyber risk evaluation engine and supports the replacement of vulnerable CPS components in manual or continuous automatic modes. While the vulnerable component replacement is in progress, our solution reconfigures the CPS to reduce the attack surface and make the CPS more resilient against cyber attacks. In our CPS reconfiguration experiment, we found that the overall process of changing the IP address of a Raspberry Pi and automatic reconnection to a new OPC UA server takes 9.7 s on average. The actual time to change the IP address of the Raspberry Pi is negligible compared with the time taken to reconnect with an OPC UA server.

## 6. Future Work

We plan on expanding our VERCASM-CPS software package into the form of virtualized and distributed Linux environments. We also plan to improve the performance of our solution by employing a caching technique and batch processing. To store configurations used in each microservice in our framework, we plan to employ a "PROSPECD" secure data container [43], which supports fine-grained role-based and attribute-based access control. This would also be integrated with a Relational Database Management System.

**Author Contributions:** Conceptualization, B.N., D.U., T.B. and M.R.; methodology, B.N., D.U., T.B. and M.H.; Software, B.N. and T.B.; validation, B.N., T.B. and M.H.; formal analysis, M.H. and D.U.; investigation, B.N., D.U., T.B. and M.H.; resources, B.N., D.U. and T.B.; data curation, B.Northern; writing—original draft preparation, B.N., D.U., T.B. and M.H.; writing—review and editing, B.N., D.U., T.B., M.H. and M.R.; visualization, B.Northern and T.B.; supervision, D.U.; project administration, D.U.; funding acquisition, D.U. All authors have read and agreed to the published version of the manuscript.

**Funding:** This research received no external funding.

**Institutional Review Board Statement:** Not applicable.

**Informed Consent Statement:** Not applicable.

**Data Availability Statement:** For evaluating our methodology, we used publicly available database of vulnerabilities at https://nvd.nist.gov/vuln/data-feeds (accessed on 22 August 2021), applied SQL query from Listing 3 and the formula from Theorem 1.

**Acknowledgments:** We thank the Department of Computer Science and Cyber Education Research and Outreach Center (CEROC) at Tennessee Technological University for providing funds and resources for this project. We thank David Smith from Tennessee Technological University for providing mathematical insight to aspects of this paper. We thank Robert Craven and Haley Smallwood from Center for Energy Systems Research (CESR) at Tennessee Technological University for help with assembling the experimental testbed with the power meter.

**Conflicts of Interest:** The authors declare no conflict of interest.

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
