# Peer review of "VERCASM-CPS: Vulnerability Analysis and Cyber Risk Assessment for Cyber-Physical Systems"

_information, doi:10.3390/info12100408_

Round 1

Reviewer 1 Report

The paper discusses an interesting topic and the authors covered the topic very well. However, I have some minor comments that could improve the papers' quality. 

  • Some technical details and numerical results must be added to the abstract to show the results of the proposed method.
  • figure 1 is very generic and I recommend adding a framework instead of using  UML in representing your idea
  • Please be specific in writing the sections' titles. For example, section 3 ( Core Design) ! core design for what? this creates some confusion for the reader. 
  • Again, avoid general discussion or general structures like UML, XML ... for example, it is not recommended to add something like 

    Listing 1: Example XML. If it is urgent, you can add it in a form of a pseudocode 

  • All other parts are Ok

Author Response

Point 1: Some technical details and numerical results must be added to the abstract to show the results of the proposed method.

Response 1: Several quantitative results of the proposed method have been added to the end of the Abstract. 

Point 2: figure 1 is very generic and I recommend adding a framework instead of using  UML in representing your idea

Response 2: Figure 1 has been modified to a workflow diagram, and the old UML has been moved to figure 2 as it explains in greater detail the system and each sub-component in our design.

Point 3: Please be specific in writing the sections' titles. For example, section 3 ( Core Design) ! core design for what? this creates some confusion for the reader.

Response 3: Section 3 title has been renamed to “VERCASM-CPS Core Design”

Point 4: Again, avoid general discussion or general structures like UML, XML ... for example, it is not recommended to add something like

Listing 1: Example XML. If it is urgent, you can add it in a form of a pseudocode

Response 4: Listing 1 has been modified to use a pseudocode instead of XML. Corresponding example description in the “API Gateway” subsection has been modified, accordingly. 

Reviewer 2 Report

The paper present an interesting approach toward Cyber physical vulnerability assessment. The solution presented is very interesting showcasing a system that can evaluate and discover CPS vulnerabilities.

It will be good to reference other work in the area such us doi: 10.3390/s21144939. Where the part of Software defined networking collaborate with a Vulnerability assessment framework.

Other work in the area that will be interested to review and reference can be:

10.1109/ICMLA.2019.00206 where deep neural network is utilised for flow based controlled. The use of deep neural network can be helpfull and can assist in the further development of the paper.

Last but not list the part of use of middleware in the edge can help in a tailored based security service in the edge an interesting approach is 10.1109/MCOM.2019.1800506 where the use of an accelerated device near the edge can assist in the deployment of cyber solutions near the edge.

In the paper errors:

Line 109 a tab is forgotten.

Line 114-115 The Phrase The authors... up to....by themselves. Is hard to follow please rephrase.

Line 121 A reference is forgotten (Tripathi and Singh)

Line 126 The reference must go after the names.

Line 130 The phrase is informal "what is know ... which are featured...

Line 170 The reference to be moved after the name.

Line 269-270 Reference missing in the part of Volatile memory and the high amount used in DBMS.

Listing 4 & 5 is redundant for a journal publication

Line 360 up to 367 to be added in a table.

Section 3.5 it will be intresting to see and reference the work in  10.1109/WF-IoT.2019.8767249 where assessment as a service is proposed as a solution for entering devices.

Last but not list the EPSS MODEL will be much more easy to be utilised as it will show in clear manner how exploitable is the vulnerability identified (https://www.first.org/epss/model)

Author Response

Point 1: It will be good to reference other work in the area such us doi: 10.3390/s21144939. Where the part of Software defined networking collaborate with a Vulnerability assessment framework.

Response 1: A paragraph with the suggested reference has been added in the “Related work” section. We compared and contrasted our solution and theirs.

Point 2:  Other work in the area that will be interested to review and reference can be:

Point 2.1: 10.1109/ICMLA.2019.00206 where deep neural network is utilised for flow based controlled. The use of deep neural network can be helpful and can assist in the further development of the paper.

Response 2.1: A paragraph with 3 sentences has been added to the “Related Work” section. It suggests the direction of how deep neural network-based IDS can be integrated with our solution to make CPS more secure. 

Point 2.2: Last but not list the part of use of middleware in the edge can help in a tailored based security service in the edge an interesting approach is 10.1109/MCOM.2019.1800506 where the use of an accelerated device near the edge can assist in the deployment of cyber solutions near the edge.

Response 2.2: A paragraph about accelerators and a potential use case with our system was added to the Related Work.

In the paper errors:

Point 3: Line 109 a tab is forgotten.

Response 3: Fixed, a new line was created between lines. 

Point 4: Line 114-115 The Phrase The authors... up to....by themselves. Is hard to follow please rephrase.

Response 4: The sentence mentioned and the one after describing the factors have been combined to make the information clearer.

Point 5: Line 121 A reference is forgotten (Tripathi and Singh)

Response 5: Citation has been added at line 122.

Point 6: Line 126 The reference must go after the names.

Response 6: Citations have been moved to after the names of all cited authors.

Point 7: Line 130 The phrase is informal "what is known ... which are featured…

Response 7: Removed these phrases from the sentence.

Point 8: Line 170 The reference to be moved after the name.

Response 8: Citations have been moved to after the names of all cited authors, for all cited papers.

Point 9: Line 269-270 Reference missing in the part of Volatile memory and the high amount used in DBMS. 

Response 9: The sentence with the missing reference has been removed. 

Point 10: Listing 4 & 5 is redundant for a journal publication

Response 10: Listings 3 and 4 have been removed. We added references back to listing 5 which is now listing 3.

Point 11: Line 360 up to 367 to be added in a table.

Response 11: We added the specified lines into table 2 to clear up confusion.

Point 12: Section 3.5 it will be interesting to see and reference the work in  10.1109/WF-IoT.2019.8767249 where assessment as a service is proposed as a solution for entering devices.

Response 12: A paragraph briefly summarizing the paper by Markakis et al. has been added to the related work. We also compare and contrast their proposed solution with our methodology.

Point 13: Last but not least the EPSS MODEL will be much more easy to be utilised as it will show in clear manner how exploitable is the vulnerability identified (https://www.first.org/epss/model)

Response 13: A paragraph with 4 sentences has been added to the “Related Work” section to review the EPSS model. It is also mentioned in subsection 3.5 of the “VERCASM-CPS Core Design” and compared with our solution for the Controlled Moving Target Defense.   

Round 2

Reviewer 1 Report

It can be accepted now

Reviewer 2 Report

The authors addressed all my comments